# Within-subject variation of C-reactive protein and high-sensitivity C-reactive protein: A systematic review and meta-analysis

**Alex Gough**📶*, **Alice Sitch, Erica Ferris, Tom Marshall**

Institute of Applied Health Research, College of Medical and Dental Sciences, University of Birmingham, Birmingham, United Kingdom

* Agg932@student.bham.ac.uk

## Abstract

### Background

C-reactive protein (CRP) and high-sensitivity C-reactive protein (hsCRP) are measures of inflammation used in diagnosis, to guide treatment decisions, and in disease prediction. Variability in measured CRP and hsCRP may affect their clinical utility but estimates of within-subject variability are based on limited data.

### Methods

A systematic review and meta-analysis was performed to estimate longitudinal within-subject variability of CRP and hsCRP over any time period. Follow-up studies of any design in adults or children, with repeated measures of CRP or hsCRP were sought. Multiple databases were searched from inception to November 2022. Titles and abstracts were screened in duplicate. Full text screening and data extraction were performed by one reviewer and verified by a second. Risk of bias was assessed with a modified Consensus-based Standards for the Selection of Health Measurement Instruments (COSMIN) tool. Intraclass correlation coefficient (ICC) results were pooled with a meta-analysis and coefficient of variation (CV) results were described by median and range.

### Results

Of 2675 studies identified, 60 met the inclusion criteria: 34 reported CRP and 26 reported hsCRP. For CRP, median CV was 0.41 (range 0.11 to 0.89), and the pooled estimate of ICC was 0.55 (95% CI 0.35 to 0.74). For hsCRP, median CV was 0.44 (range 0.27 to 0.76) and the pooled estimate of ICC was 0.62 (95% CI 0.58 to 0.67).

### Limitations

Assessment of variability was not the main aim of many of the included papers, and it is possible that some relevant papers have been missed. Many of the papers included had low numbers of participants and/or low numbers of repeated measurements.

**Data Availability Statement:** All data are in the manuscript and/or Supporting information files.

**Funding:** The author(s) received no specific funding for this work.

**Competing interests:** The authors have declared that no competing interests exist.

## Conclusions

Estimated within-subject variability is high for both CRP and hsCRP, but estimates are based on small numbers of participants and measurements. There is a need for better estimates of within-subject variability from analysis of larger numbers of repeated measurements in larger numbers of subjects.

## Introduction

C-reactive protein (CRP) is a non-specific biomarker for acute inflammatory diseases such as infections (including COVID-19) and for chronic inflammatory conditions such as cardiovascular diseases [1]. Plasma levels may increase up to a thousand-fold in the presence of inflammation and tissue trauma [2]. Mild, chronic elevations are associated with an increased risk of cardiovascular disease [3]. Point of care (POC) CRP can be used to aid the diagnosis and inform antibiotic treatment of respiratory tract infections (LRTIs) [4]. CRP is therefore used to inform diagnosis, prognosis and aid in therapeutic decisions making.

The CRP test evaluates a higher range of values than hsCRP. It is used to diagnose acute inflammatory conditions such as infection and to diagnose or monitor chronic inflammatory conditions like rheumatoid arthritis. The hsCRP test uses immunoassay methods which are sufficiently precise to quantify CRP throughout its normal range [3]. hsCRP helps predict risk of cardiovascular disease. Its most important use is to identify patients with known atherosclerotic cardiovascular disease with residual inflammatory risk [5].

Various cut-off points are used when using CRP as an aid to diagnosis. A value of 0 to 0.3mg/L is considered normal, 1-10mg/L suggests moderate inflammation (autoimmune/inflammatory diseases), 10-50mg/L suggests marked inflammation such as acute infection and more than 50mg/L suggests bacterial infection. A value less than 10mg/L suggests acute infection is unlikely [6]. Different researchers recommend different exact cut-offs. For example some suggest less than 10mg/L is normal and above 10mg/L indicates pathology [7]. NICE guidelines advise that bacterial lower respiratory tract infection is unlikely with a CRP <20mg/L [8]. Although the reference range for hsCRP is considered to be 0 to 0.3 mg/L, this may vary in different settings. Higher average values have been observed in healthy adults in both Iran and South Korea [9, 10].

Reference ranges for diagnostic tests are usually derived from a reference population, which allows assessment of between-subject variation. However, biological variables also vary within individuals over time. This can be systematic and predictable, such as circadian or seasonal variation, or may be due to chance. This type of longitudinal within-subject variation ($CV_I$) is known as biological variation. Variation can also be introduced into a measurement from pre-analytical factors. For example, transient infections, physical activity and stress increase CRP levels [11, 12] and CRP also increases with age [13]. Imperfect accuracy and precision of a test suggest that if a test is repeated a number of times, there will be a range of results around the true value. This is analytical variation ($CV_A$). Analytical and biological within-subject variation, $CV_A$ and $CV_I$, together with any pre-analytical factors affecting the measurement, combine to give the total within-subject variation ($CV_T$). The coefficient of variation (CV), which is the standard deviation divided by the mean, is the most common statistic used to describe within subject variation, but various other statistics are sometimes used, especially standard deviation and intraclass correlation (ICC).

Measured CRP results may differ from the true mean, i.e. the real mean value of the parameter in an individual. This can be due to a combination of biological, pre-analytical and

analytical variability. This total variability is the variability encountered in real-world clinical practice. Biological variability is often the greatest component of total variability, since clinical laboratories usually set acceptable analytical variability at <0.5% of biological variability [14]. Different clinical parameters have different levels of biological variability. For example, one study estimated the within-subject coefficient of variation ($CV_i$) of creatinine as 6.4% in people without chronic kidney disease [15] and another study estimated the $CV_i$ of glucose in non-diabetics as approximately 5% [16].

CRP is considered to have a high biological variability. The European Biological Variability Database lists the $CV_i$ of CRP as 34.1% (95%CI 29.4–74.4) and hsCRP as 58.9% (53.2–66.0) [17]. It has been argued however that there may be inaccuracies in published reports, for example because of inclusion of subjects with transient illness. In theory values higher than 33% for $CV_i$ should be treated with suspicion [11].

The greater the within-subject variability of a parameter, the lower the probability that a single measurement is an accurate reflection of the subject's true mean. This can lead to errors in diagnosis, prognosis and treatment. Specifically for CRP, an inaccurate result may lead to inappropriately prescribing or withholding antibiotics in cases of sepsis and respiratory tract infections. An inaccurate hsCRP result may give inaccurate diagnoses and risk assessments for cardiac diseases. It has been shown that CRP values affect clinical decision making. When paediatricians were given case vignettes with identical radiographs and clinical data, but randomised so that they were given CRP values either low (23mg/L) or high (151mg/L), they were 60–90% more likely to diagnose bacterial pneumonia rather than viral pneumonia and consequently to prescribe antibiotics [18].

Some authors even suggest the level of variability of CRP may influence clinical decisions, with one paper studying intensive care unit acquired infections showing daily CRP infection could be used as a marker of infection [19].

Although there have been a number of studies on the biological variability of CRP and hsCRP, there are no large-scale meta-analyses or systematic reviews. The meta-analyses on the European Biological Variability Database [17] are based on only four papers for CRP and one paper for hsCRP.

## Aim

To describe and evaluate the current literature on the within-subject variability of CRP and hsCRP in healthy subjects and patients with medical conditions and to estimate within-subject variability of these measures.

## Methods

Searches were devised to identify cohort studies, clinical trials or any studies in which CRP or hsCRP was measured more than once in the same individual. See Appendix 1 in S1 File for Medline and Embase search strategy. The searches for this study were combined with a similar systematic review of HbA1c variability, and then HbA1c studies were excluded at the full text screening stage.

Studies were included if primary research data on the variability of at least two measurements of CRP or hsCRP within the same subject was recorded. Studies could be of any design. The population included adults and children, healthy or with any disease condition, in any setting. Outcome was variability which could be reported as coefficient of variation ($CV_I$), standard deviation (SD), variability independent of the mean (VIM), index of individuality (II), Reference Change Value (RCV), index of heterogeneity, validity coefficient (VC), ICC agreement, ICC consistency, Cronbach's alpha or Cohen's kappa. There was no restriction on time

of publication, language of publication, population, setting or sample size. Studies were excluded if participants were not in a steady state (measurements were before and after an intervention or had an acute or rapidly changing illness) or data were secondary (systematic and narrative reviews).

Medline, Embase, Cochrane Central, Epistemonikos and Open Grey were searched from inception to 5[th] August 2020. An update to the search was performed to include studies published up to November 2022 (Full details in Appendix 2 in S1 File). Search terms were adapted for each database searched. Subject experts were contacted for suggestions for further papers. The references of included papers were checked by hand for further relevant papers. PROSPERO was checked for ongoing reviews, and the protocol was registered with PROSPERO. Endnote reference management software was used to collate studies revealed by the search.

Titles and abstracts were screened independently by two reviewers in Abstrackr systematic review software [20], and full texts were obtained for those meeting inclusion criteria where possible. Full texts were screened by AG and exclusions confirmed by EF. Differences were resolved by discussion. Foreign language papers were translated by Google translate software.

If full texts met the inclusion criteria, a single reviewer (AG) extracted data into Excel from the full text, or from abstracts where full texts were not available. All data extraction was verified independently by a second reviewer (EF). Where the same study was reported in multiple papers, the full text paper was preferred over an abstract, English language preferred over non-English, and the earliest English version over the later if there was more than one.

Table 1 lists the outcome and other main variables extracted. All eligible outcomes were included where more than one outcome was reported within a paper. The primary outcome measure was variability of repeated measurements within the same subject. Where multiple measures of variability were given in a single study, the primary population only was analysed for the main meta-analysis. The primary population was the full study population (as opposed to subgroups), and if the full study population was not given, the primary outcome was identified using the following hierarchy: 1. Healthy population; 2. Most stable population (i.e.

**Table 1. Variables extracted from papers.**

| Variable | Definition and rules |
|---|---|
| CRP or hsCRP | Whether high sensitivity CRP was specified in the text |
| Study design | Cohort, RCT etc |
| Number of subjects | Number of subjects used to calculate variability measure |
| Age | Average age of subjects used to calculate variability measure. If the subjects used to calculate variability measure is a subset of the study population and only the age is given for the whole population, the age of the whole population is used. |
| Sex | Percentage of subjects used to calculate variability measure who were male. If the subjects used to calculate variability measure is a subset of the study population and only the sex is given for the whole population, the sex of the whole population is used. |
| Ethnicity | Ethnicity recorded yes/no? If country of origin of subjects only is recorded, this is counted as no. |
| Setting | Primary care/community versus secondary/tertiary/laboratory setting |
| Health status | Healthy v inflammatory disease v cardiac disease v haemodialysis v other |
| Number of measurements | Number of repeated CRP measurements in the same individual |
| Time interval between measurements | Length of time between measurements in days. |
| CVI | Coefficient of variation of repeated measures within an individual |
| SD | Standard deviation of repeated measures within an individual |
| ICC | Intraclass correlation of repeated measures within an individual |

subjectively judged to be in the most steady-state such as disease course or treatment); 3. First outcome listed in the paper. If both CRP and hsCRP were included, the data were extracted separately for each measurand.

Where information was missing or unclear, the information was not extracted with the following exception: where patient characteristic data was reported only for a whole study population, but variability data was only present for a sub-population, the patient characteristic data for the whole population was used.

A risk of bias tool was adapted from the COSMIN tool for test reliability [21]. It included seven questions regarding patient stability, time between measurements, differences in measurement conditions, administering the measurement and assigning scores without knowledge of previous scores, any other design flaws, and whether the variability measure was adequately described. These were rated on a four-point scale from very good to inadequate. Risk of bias was scored by AG and the scores were reviewed by EF. Disagreements were resolved by discussion. Risk of bias was also assessed using the BIVAC method [22]. See Appendix 3 in S1 File.

Studies were grouped for synthesis based on test (CRP vs hsCRP) and variability measure reported eg ICC, CV. hsCRP and CRP measurements were analysed separately. Subgroup analyses by health status of subpopulations, setting, unit of measurement and long versus short term measure of variability were performed. Coefficients of variation were converted from % to a decimal fraction. Units of CRP measurement were converted to mg/L.

Stata SE 17 was used to perform statistical calculations and generate forest plots (see Appendix 4 in S1 File). ICCs were transformed using Fisher's Z transformation and 95% confidence intervals were calculated [23]. Z scores were then back-transformed (See Appendix 4 in S1 File). Coefficient of variations were considered for meta-analysis, but no well-described and validated methods currently exist for meta-analysis of coefficient of variations, so we describe the results identified across studies. Sensitivity analyses were carried out based on number of participants and risk of bias.

## Results

After database searches and hand searching of reference lists of included studies, 2675 non-duplicate studies were retrieved, and 2320 studies (86.7%) were excluded after title and abstract screening leaving 355 (13.3%) articles (see Fig 1). Studies were excluded at the title and abstract screening stage if it was clear there was no information on variability of either HbA1c or CRP in the study. After full text screening, 295 (83.1% of full texts screened) studies were excluded. Of these, 109 (36.9%) were excluded from this study because they reported variability for HbA1c only and not CRP, 162 (54.9%) were excluded because they did not report any variability data, 10 (3.4%) were excluded because they did not report any primary data (eg they were a review study), 10 (3.4%) were excluded because they were a duplicate paper or population, and 1 (0.3%) was excluded because it reported a non-standard measurement of variability (the mean absolute residual around the line connecting index value with closing value).

Sixty (16.9% of full texts screened) studies met the inclusion criteria, including between four and 56,218 subjects. Fifty-four studies (90% of included studies) were of a cohort design and six (10%) used data from the placebo arm of a randomised controlled trial. Fourteen studies (23.3%) were carried out in a primary care or community setting and 38 (63.3%) studies were carried out in secondary or tertiary setting such as universities, hospitals and laboratories. In eight (13.3%) studies the setting was not reported. The full reference list for these studies can be found in the supporting information.

Thirty-four (56.7%) papers reported CRP results, and 26 (43.3%) reported hsCRP results. Fifty-one (84%) papers reported long term variability (arbitrarily defined as 7 or more days

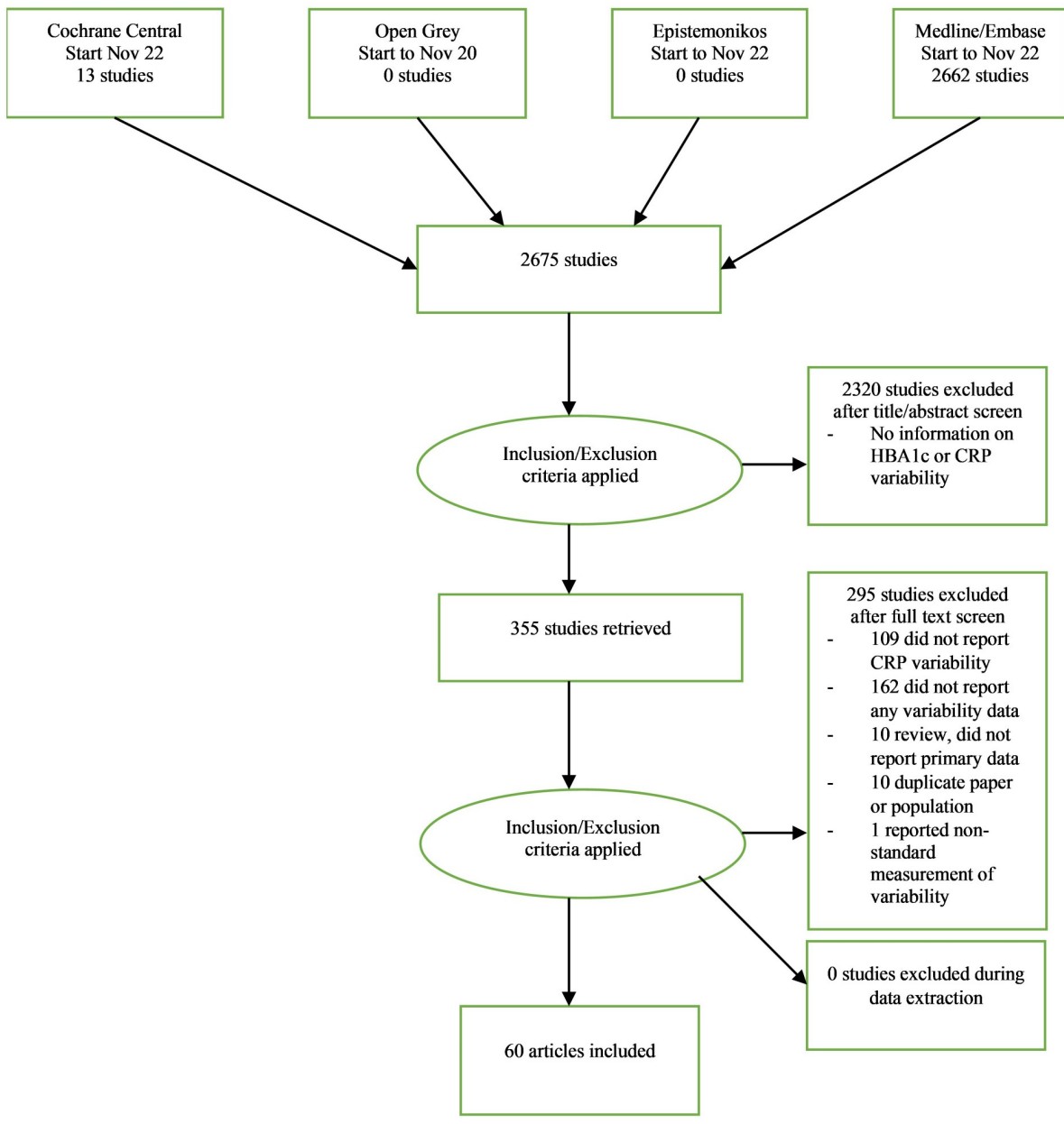

**Fig 1. PRISMA flow diagram describing selection of studies.**

between measurements), six (10%) reported short term variability (less than 7 days between measurements), two papers reported both long and short term, and one did not report measurement intervals. Study populations were diverse in terms of age, gender and health status. Ethnicity was recorded in 11 (18.3%) papers. In 22 (36.7%) studies the within-subject biological variation ($CV_I$) was reported, while in 35 (58.3%) studies the within-subject total variation ($CV_T$) was reported. In three (5.0%) studies it was uncertain whether total or biological variation was reported. For CRP, the number of studies reporting $CV_I$, $CV_T$ or it was uncertain were 14/34 (41.2%), 18/34 (52.9%) and 2/34 (5.9%) respectively. For hsCRP the number of studies reporting $CV_I$, $CV_T$ or it was uncertain were 8/26 (30.7%), 17/26 (65.4%) and 1/26 (3.8%) respectively.

**Table 2. Summary statistics of study characteristics and results of included papers.**

| Study characteristics | Number of observations | Median | IQR | Min | Max |
|---|---|---|---|---|---|
| Number of subjects | 59 | 50 | 139 | 4 | 56218 |
| Age (years) | 40 | 49.5 | 33.09 | 18 | 72.86 |
| % male | 51 | 50 | 42 | 0 | 100 |
| Number of measurements | 58 | 4 | 6 | 2 | 18 |
| **Study results** | | | | | |
| CV | 32 | 0.421 | 0.183 | 0.106 | .894 |
| SD | 7 | 0.8 | 1.93 | 0.035 | 2.04 |
| ICC | 19 | 0.62 | 0.2 | 0.174 | 0.9 |

Table 2 shows summaries of the study characteristics and results. See Appendices 5 and 6 in S1 File for full characteristics of included papers.

For CRP there were 21 studies (35.0% of all studies) that reported a $CV_I$, with a range of 0.11 to 0.89 and a median of 0.41. For hsCRP there were 11 studies (18.3%) that reported a $CV_I$ with a range of 0.27 to 0.76 and a median of 0.44. For CRP there were seven studies that reported an ICC. The pooled estimate calculated in this study for CRP ICC was 0.55 (95% CI 0.35 to 0.74). For hsCRP there were 12 papers that reported an ICC. The pooled estimate calculated in this study for hsCRP ICC was 0.62 (95% CI 0.58 to 0.67).

Figs 2 and 3 show forest plots of all papers that reported a $CV_I$ split by health status and sorted by number of subjects for CRP and hsCRP respectively. Results were similar for the healthy and unhealthy groups. Figs 4 and 5 show forest plots with meta-analysis of all papers that reported an ICC, split by health status and sorted by number of subjects for CRP and hsCRP respectively. Again there is little difference between the groups.

Figs 6 to 9 in Appendix 7 in S1 File show forest plots of CVs for CRP categorised by risk of bias, setting, short or long term variability and unit of measurement respectively. Figs 10 to 12

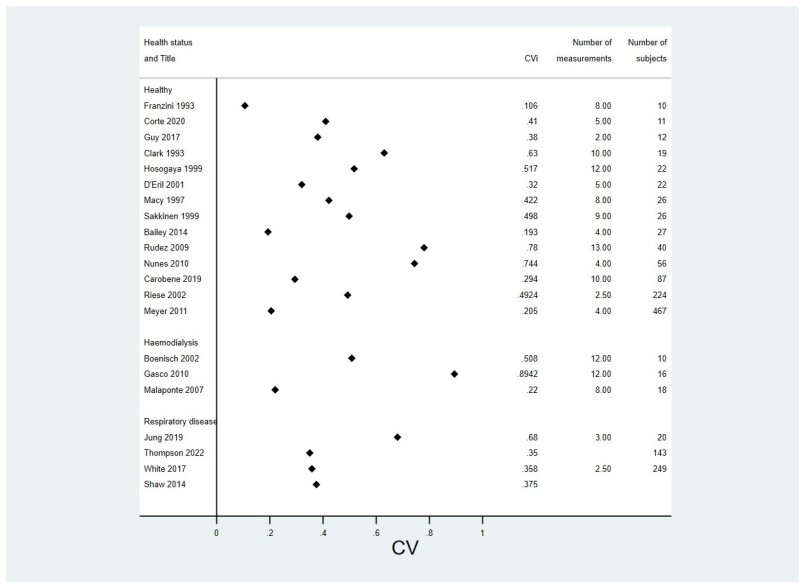

**Fig 2. Modified forest plot of all papers that reported a CV for CRP, split by health status.**

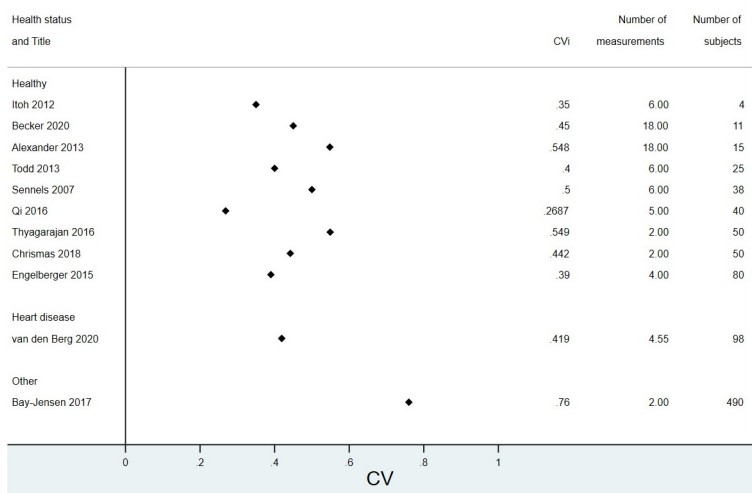

**Fig 3. Modified forest plot of all papers that reported a CV for hsCRP, split by health status.**

in Appendix 7 in S1 File show forest plots of CVs for hsCRP categories by risk of bias, setting and short or long term variability respectively. Figs 13 and 14 in Appendix 7 in S1 File show forest plots with meta-analysis categorised by risk of bias for ICCs of CRP and hsCRP respectively. There was little difference between the subgroups. See Appendix 7 for Figs 6 to 14 in S1 File.

When assessing risk of bias, no study scored more than doubtful for question B4 *knowledge of previous scores when administering the test* and B5, *when assigning the scores (questions B4*

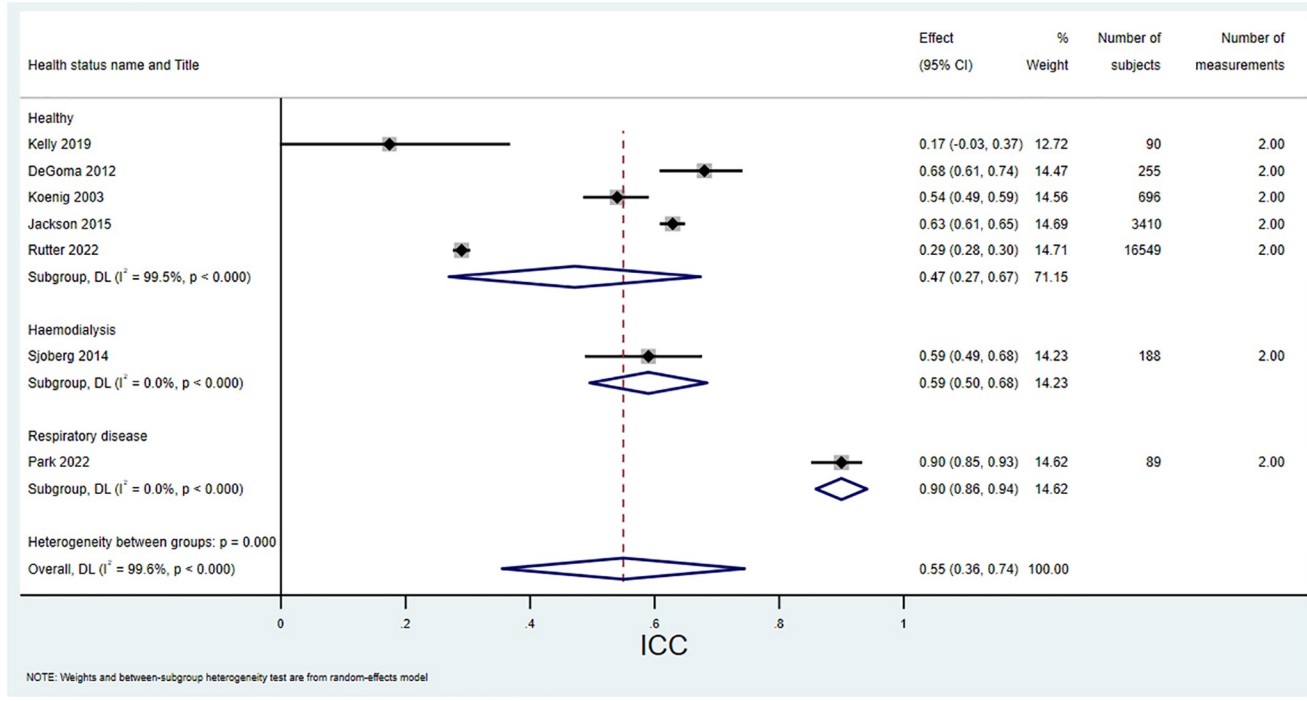

**Fig 4. Forest plot of all papers that reported an ICC for CRP, split by health status, with meta-analysis.** Note that the lower confidence interval for Kelly was manually truncated at 0, since the meta-analysis gave a negative value which is meaningless for an ICC.

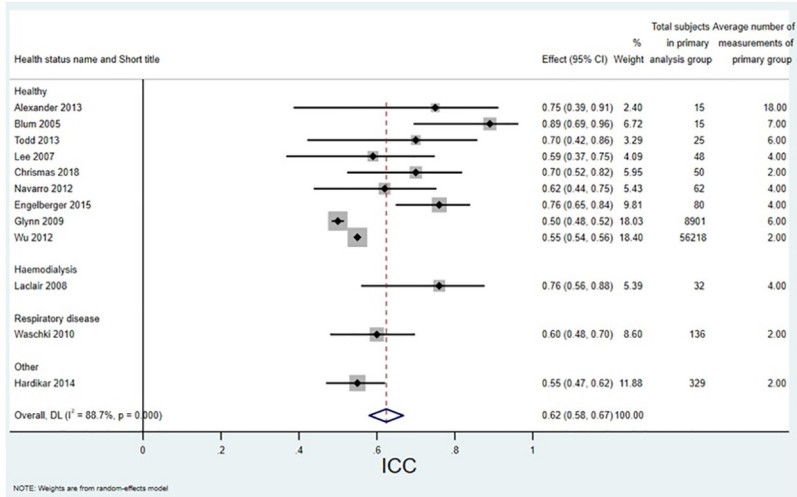

**Fig 5. Forest plot of all papers that reported an ICC for hsCRP, split by health status, with meta-analysis.**

and B5). Table 3 shows the frequency of the maximum risk of bias with questions B4 and B5 excluded since including them would mean that all studies would be rated at a high risk of bias. The inclusion criteria ruled out inadequate studies.

Risk of bias due to missing results was deemed to be low, since in most cases the outcome measure of this review was not the primary outcome of the individual studies, and therefore withholding publication based on unwanted results is unlikely. Some papers with repeated measures do not report variability data, but this is likely to be because it is not of relevance to the study. As such this should happen at random, rather than introducing a systematic bias. If studies with missing results had missing data systematically related to the variability measures, this could introduce bias, but there is no reason to think this is the case.

## Discussion

This review describes the current literature on variability of depression measurement instruments within an individual. While it is recognised that CRP has high biological variability, the data regarding this is spread throughout the literature, with the only meta-analysis on the subject based on very low numbers of studies. This current systematic review and meta-analysis has a more transparent and consistent search strategy than the existing database. It includes a much wider pool of studies, and is not restricted to healthy populations, allowing some conclusions to be drawn on the variability of CRP and hsCRP in unhealthy populations. Studies reported a wide range of variability.

**Table 3. Highest risk of bias with questions B4 and B5 excluded (1 is best possible score, 4 is worst possible score).**

| Maximum risk of bias | Number of studies (%) |
|---|---|
| 1 | 38 (63.3%) |
| 2 | 15 (25%) |
| 3 | 7 (11.7%) |
| 4 | 0 (0%) |

For CRP, the median CV was 0.41 with a range of 0.11 to 0.89 and the pooled estimate ICC was 0.55 (95% CI 0.35 to 0.74). For hsCRP the median CV was 0.44 with a range of 0.27 to 0.76 and the pooled estimate ICC was 0.62 (95%CI 0.58 to 0.67). These compare with the pooled results for $CV_I$ given on the European Federation of Clinical Chemistry and Laboratory Medicine (EFLM) Biological Variation Database [17] of 0.34 (95%CI 0.29 to 0.74) for CRP and 0.59 (95%CI 0.53 to 0.66) for hsCRP. This likely reflects a wide range of health statuses, settings, time between measurements and number of measurements. The ELFM database [17] largely includes studies on healthy subjects. However, subgroup analyses and sensitivity analyses in this study showed little difference between subgroups.

The paper by Kelly [24] is an outlier with no obvious explanation for the very low ICC. The paper also examined within-subject variability of HbA1c and reported a much higher ICC for HbA1c of 0.873. The study was unremarkable in terms of setting and patient characteristics. However, the paper was a conference abstract with no full text, so further detail and discussion of this result was lacking.

There is no standard agreement for what level of coefficient of variation or intraclass correlation constitutes "low" or "high" variability, and in fact this needs to be individualised for each parameter. A measurand that has a large within individual coefficient of variation, but a large disparity in measured values between healthy and diseased populations may have less clinical implication than a lower level of variation but a lower disparity between healthy and diseased values. Nevertheless, the level of variability for CRP and hsCRP shown in this systematic review is subjectively high.

In theory the CV of a positive normally distributed set of data should not exceed 33.3% (11) which suggests that either the range of values of CRP is not normally distributed or that some of the values are incorrect due for example to transitory illnesses. However, there seems to be sufficient data in the current literature to suggest that whatever the "true" figure for CRP variability, it is high enough to be clinically significant. This is particularly true when single number cut-off points are used, for example "do not use antibiotics if CRP <20mg/L", since a measured value of for example 18mg/L may represent a much higher (or much lower) true mean for that individual. While CRP alone is not the basis for clinical decisions on treatment with antibiotics, since many factors besides infection can affect CRP levels, if using CRP measurement results to aid diagnostic decisions, clinicians should be aware that a single measurement, or even multiple measurements, may not be an accurate reflection of the true mean CRP for an individual patient. Within-individual variability may also be important with regard to hsCRP levels, particularly in the setting of cardiovascular disease.

The papers included in this study often had low numbers of participants and low numbers of repeated measurements. The studies with high numbers of participants tended to have low numbers of repeated measurements and vice versa, likely for practical and cost reasons. There is thus a lack of studies that show a large number of repeated measurements in a large population. None of the studies were blinded, (it was not ensured that the professionals administered the tests and assigned the scores without knowledge of previous patient scores).

There are some limitations with this review. Not all studies that report variability data have the primary aim of measuring variability, so it is possible that some studies with data on CRP variability may have been missed. However, a broad range of search terms was employed which retrieved a large number of titles and abstracts, which in turn led to a large number of full texts being screened for inclusion.

For meta-analysis, only a small number of studies could be pooled because of different tests being performed (CRP v hsCRP), different variability measurements being calculated (CV, SD and ICC) and lack of well-described validated methods for performing meta-analysis of coefficients of variation. It should also be noted that some papers reported total within-subject

variability and some biological within-subject variability, and in some cases an assumption was made as to which of these had been calculated. In a few cases it was not even possible to make an assumption as to which type of variability was being described (see Appendix 5 in S1 File).

The inclusion of data from the placebo arm of randomised controlled trials could be considered to be problematic if placebo has an effect on the variability. However, the authors are not aware of any evidence that this effect can occur.

Further studies are required that have both larger populations and more repeat measurements, in a variety of different health statuses, to enable the variability of CRP and hsCRP to be better described. Future research should consider standardisation of reporting of measures of variability such as using the coefficient of variation, and making sure that the values for within individual variation are clearly reported. Standardised biological variation studies such as those listed in the ELFM database generally use healthy populations, but it is important that research is directed at ascertaining the biological variability in populations with various disease statuses. The implications of within individual biological variation on reference ranges, the formulation of clinical guidelines and clinical decision making need to be investigated.

Nevertheless, this systematic review and meta-analysis provides new information on the within-subject variability of CRP. The high variability shown by this review demonstrates the need for clinicians and producers of guidelines to carefully consider how accurately a single CRP result reflects the individual's true CRP level, and how many repeat tests may be needed to provide an accurate result.

## Other information

### Registration

CRD42020202387 Registered 17/09/2020.

The review protocol is available at https://www.crd.york.ac.uk/prospero/display_record.php?ID=CRD42020202387.

The protocol describes a risk of bias tool which was adapted from the QUADAS tool. The study used the COSMIN tool for risk of bias assessment as being more appropriate and validated. The original study intended to include HbA1c and CRP in a single paper, but for reasons of size and to improve targeting of appropriate outlets for dissemination, the study was separated into HbA1c and CRP after the data extraction stage.

## Supporting information

**S1 Checklist. PRISMA checklist.**
(DOCX)

**S1 File. Appendices 1 to 7.**
(DOCX)

## Author Contributions

**Conceptualization:** Alex Gough, Alice Sitch, Tom Marshall.

**Data curation:** Alex Gough, Erica Ferris.

**Formal analysis:** Alex Gough, Erica Ferris.

**Investigation:** Alex Gough.

**Methodology:** Alex Gough, Alice Sitch, Tom Marshall.

**Project administration:** Alex Gough.

**Writing – original draft:** Alex Gough, Alice Sitch, Tom Marshall.

**Writing – review & editing:** Alex Gough, Alice Sitch, Tom Marshall.

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
