## [Decision Letter · Decision Letter 0]

13 Mar 2024

PONE-D-23-37309Within-subject variation of C-reactive protein and high-sensitivity C-reactive protein: A systematic review and meta-analysisPLOS ONE

Dear Dr. Gough,

Thank you for submitting your manuscript to PLOS ONE. After careful consideration, we feel that it has merit but does not fully meet PLOS ONE’s publication criteria as it currently stands. Therefore, we invite you to submit a revised version of the manuscript that addresses the points raised during the review process in a detailed fashion.

We look forward to receiving your revised manuscript.

Kind regards,

Andreas Zirlik, MD

Academic Editor

PLOS ONE

Journal Requirements:

Reviewers' comments:

Reviewer's Responses to Questions

**Comments to the Author**

1. Is the manuscript technically sound, and do the data support the conclusions?

Reviewer #1: Partly

Reviewer #2: Partly

2. Has the statistical analysis been performed appropriately and rigorously? 

Reviewer #1: I Don't Know

Reviewer #2: I Don't Know

3. Have the authors made all data underlying the findings in their manuscript fully available?

Reviewer #1: Yes

Reviewer #2: Yes

4. Is the manuscript presented in an intelligible fashion and written in standard English?

Reviewer #1: Yes

Reviewer #2: Yes

5. Review Comments to the Author

Reviewer #1: Review

Authors performed a meta analysis of studies reporting variability in different populations. They report large intraindidivual variability over time. While this is interesting to clinicians (potential overuse of CRP), there are major concerns regarding the design of the meta analysis and the quality of the underlying studies.

Abstract

Line 28-29: A systematic review and meta-analysis was performed to estimate within-subject variability of CRP and hsCRP. � Please add, that variability over time was measured and state which time period was considered.

Introduction

Line 70: while it may be true that hsCRP can predict the risk of cardiovascular disease I do not agree that this it is used for this purpose in the clinic often. I would say that the most important use of hsCRP is to identify patients with known atherosclerotic cardiovascular disease with residual inflammatory risk.

Line 95: I do not understand the sentence – “duetocause”?

Line 118-121: What are the authors trying to say? Please rephrase.

Methods:

Line 137-142: I would suggest a statistician to check the methodology as I myself am not an expert in statistics.

Line 143-144: “There was no restriction on time of publication, language of publication, population, setting or sample size”. Overall, this is problematic. This design limits the validity as biological variability will differ between populations and larger sample sizes are less likely to report random results. For example, including a study with 4 patients in the analysis is questionable.

What were the intervals between CRP measurements?

Were the studies included conducted specifically to assess CRP/hsCRP variability? If not, how can they ensure that intraindividual variability was measured and not treatment effects or disease states (trauma, infection, cancer etc.) were measured? The authors need to explain how the results they report are meaningful rather than statistical “noise”.

Line 154: Foreign language papers were translated by Google translate software. – this is problematic as it can introduce mistakes.

Results:

Regarding Line 222-223: CRP values will naturally vary over time, particularly if the study features clinical trials, as interventions will affect CRP levels (e.g. statin treatment). Later the authors state that CRP levels measured before and after interventions led to exclusion of the study. There were six clinical trials included (placebo). However, even the placebo may theoretically affect CRP levels (patient may behave differently if he believes being treated). I do not believe these studies should be included. I believe answering this question requires more controlled conditions and a prospective cohort study design.

Discussion:

The discussion reads as a long list of limitations.

Line 300: This current systematic review and meta-analysis includes a much wider pool of studies, and is not restricted to healthy populations, allowing some conclusions to be drawn on the variability of CRP and hsCRP in unhealthy populations.

Line 329: This is particularly true when single number cut-off points are used, for example “do not use antibiotics if CRP <20mg/L”, since a measured value of for example 18mg/L may represent a much higher (or much lower) true mean for that individual. Clinicians should be aware that a single measurement, or even multiple measurements, may not be an accurate reflection of the true mean CRP for an individual patient. – From a clinician’s perspective this statement is not required. Nobody would start antibiotics based alone on CRP. Please rephrase. In fact, many clinicians do not use CRP at all when assessing need for CRP. I believe that many factors influence CRP increase which is the more significant clinical problem, rather than a individual variability. This should be commented on in the discussion.

Individual variability may however be more important regarding hsCRP levels, particularly in the setting of cardiovascular disease. Authors should discuss this.

Reviewer #2: At first I would like to thank the editor for the opportunity to review the paper of Gough et al.

This paper is a meta-analysis about the estimation of within-subject variability of

CRP and hsCRP measurement resulting in a respectively high variability for both measurements, quite similar to an existing meta analysis. The study is limited by different variability measurements being calculated and a partially low participant number in the included studies. The aim and the methods used in the study were outlined clearly.

Please find the specific comments below.

1.) Impact. The authors should outline the added value of this study, especially compared to the existing, updated meta-analyses of the EFLM database, including different health conditions. What additional knowledge do the authors' results provide and how should they be interpreted in comparison to the results of the existing meta-analysis?

2.) Readability and Typography. The study would profit from breaking down the main information in shorter sentences. Authors should as well remain consistent in the way they write out numbers or describe them as digits. Some parts of the study should be proofread for easier understanding (for example lines 46, 95…)

3.) Consistency in reporting results. The study would improve if the results were pointed out clearer.

The authors should clarify the following:

a) Report of all possible scores (1-4) of “risk of bias” in table 3.

b) The authors report “In 22 (36.7%) studies the within-subject biological variation (CVI) was reported, while…”. Opposing this, about 32 observations of CV are mentioned later on, as well as in Figure 2 and 3.

It would be helpful to point out how many studies in which group (CRP vs hsCRP) were reporting CVI, CVT or CV.

6. PLOS authors have the option to publish the peer review history of their article (what does this mean?). If published, this will include your full peer review and any attached files.

Reviewer #1: No

Reviewer #2: No

---

## [Author Response · Author response to Decision Letter 0]

29 Mar 2024

Reviewer #1: Review

Authors performed a meta-analysis of studies reporting variability in different populations. They report large intraindidivual variability over time. While this is interesting to clinicians (potential overuse of CRP), there are major concerns regarding the design of the meta-analysis and the quality of the underlying studies.

Thank you for the comment. One of the aims of the systematic review is to identify studies of intra-individual variability and characterise their quality.

Abstract

Line 28-29: A systematic review and meta-analysis was performed to estimate within-subject variability of CRP and hsCRP. � Please add, that variability over time was measured and state which time period was considered.

We have clarified that by within-individual variability we mean variability over time within the same individual. There were no time restrictions on the interval between measurements within the same individual. Line 28

Introduction

Line 70: while it may be true that hsCRP can predict the risk of cardiovascular disease I do not agree that this it is used for this purpose in the clinic often. I would say that the most important use of hsCRP is to identify patients with known atherosclerotic cardiovascular disease with residual inflammatory risk.

Line 95: I do not understand the sentence – “duetocause”?

Thank you for this comment. We have revised this sentence.

Line 118-121: What are the authors trying to say? Please rephrase.

Thank you for this comment. We have revised this sentence.

Methods:

Line 137-142: I would suggest a statistician to check the methodology as I myself am not an expert in statistics.

Alice Sitch (co-author) is a statistician.

Line 143-144: “There was no restriction on time of publication, language of publication, population, setting or sample size”. Overall, this is problematic. This design limits the validity as biological variability will differ between populations and larger sample sizes are less likely to report random results. For example, including a study with 4 patients in the analysis is questionable.

We agree that variability will differ between populations and that random error may be smaller in larger studies. However no previous systematic review or meta-analysis of within-individual variability in CRP of this size or methodology has been carried out. We therefore first intended to identify all available studies and then comment on differences between studies.

What were the intervals between CRP measurements?

See response above. All intervals were allowed. 

Were the studies included conducted specifically to assess CRP/hsCRP variability? If not, how can they ensure that intraindividual variability was measured and not treatment effects or disease states (trauma, infection, cancer etc.) were measured? The authors need to explain how the results they report are meaningful rather than statistical “noise”.

Already mentioned in lines 145-147. Few studies were conducted specifically to assess within-individual variability, but the purpose of this review was to include as many studies as possible that reported within-individual variability. Studies were excluded where there was an intervention between measurements or an unstable disease state. 

Line 154: Foreign language papers were translated by Google translate software. – this is problematic as it can introduce mistakes.

Translation software was used mainly to identify whether a paper was eligible for inclusion. The only foreign language paper included was Franzini, and the figures in this are also included in a table in an English language review by Braga and Panteghini ( DOI: 10.1016/j.cca.2012.04.010)

Results:

Regarding Line 222-223: CRP values will naturally vary over time, particularly if the study features clinical trials, as interventions will affect CRP levels (e.g. statin treatment). Later the authors state that CRP levels measured before and after interventions led to exclusion of the study. There were six clinical trials included (placebo). However, even the placebo may theoretically affect CRP levels (patient may behave differently if he believes being treated). I do not believe these studies should be included. I believe answering this question requires more controlled conditions and a prospective cohort study design.

Studies with measurements taken before and after an intervention are excluded. We are not aware of any evidence that placebo affects within-individual variability, and we have mentioned this in lines 353-355. 

Line 329: This is particularly true when single number cut-off points are used, for example “do not use antibiotics if CRP <20mg/L”, since a measured value of for example 18mg/L may represent a much higher (or much lower) true mean for that individual. Clinicians should be aware that a single measurement, or even multiple measurements, may not be an accurate reflection of the true mean CRP for an individual patient. – From a clinician’s perspective this statement is not required. Nobody would start antibiotics based alone on CRP. Please rephrase. In fact, many clinicians do not use CRP at all when assessing need for CRP. I believe that many factors influence CRP increase which is the more significant clinical problem, rather than a individual variability. This should be commented on in the discussion.

Individual variability may however be more important regarding hsCRP levels, particularly in the setting of cardiovascular disease. Authors should discuss this.

These points are addressed in lines 332-338

Reviewer #2: At first I would like to thank the editor for the opportunity to review the paper of Gough et al.

1.) Impact. The authors should outline the added value of this study, especially compared to the existing, updated meta-analyses of the EFLM database, including different health conditions. What additional knowledge do the authors' results provide and how should they be interpreted in comparison to the results of the existing meta-analysis?

Our search strategy is more consistent and transparent than that of the EFLM database, and includes a larger number of studies. Lines 298-304

2.) Readability and Typography. The study would profit from breaking down the main information in shorter sentences. Authors should as well remain consistent in the way they write out numbers or describe them as digits. Some parts of the study should be proofread for easier understanding (for example lines 46, 95…)

Lines 46 and 95 corrected, manuscript proof read and sentences shortened. 

3.) Consistency in reporting results. The study would improve if the results were pointed out clearer.

The authors should clarify the following:

a) Report of all possible scores (1-4) of “risk of bias” in table 3.

Table 3 revised. 

b) The authors report “In 22 (36.7%) studies the within-subject biological variation (CVI) was reported, while…”. Opposing this, about 32 observations of CV are mentioned later on, as well as in Figure 2 and 3.

It would be helpful to point out how many studies in which group (CRP vs hsCRP) were reporting CVI, CVT or CV.

Information added in lines 235-248

---

## [Decision Letter · Decision Letter 1]

22 May 2024

Within-subject variation of C-reactive protein and high-sensitivity C-reactive protein: A systematic review and meta-analysis

PONE-D-23-37309R1

Dear Dr. Gough,

We’re pleased to inform you that your manuscript has been judged scientifically suitable for publication and will be formally accepted for publication once it meets all outstanding technical requirements.

Kind regards,

Andreas Zirlik, MD

Academic Editor

PLOS ONE

Additional Editor Comments (optional):

Reviewers' comments:

Reviewer's Responses to Questions

**Comments to the Author**

1. If the authors have adequately addressed your comments raised in a previous round of review and you feel that this manuscript is now acceptable for publication, you may indicate that here to bypass the “Comments to the Author” section, enter your conflict of interest statement in the “Confidential to Editor” section, and submit your "Accept" recommendation.

Reviewer #1: (No Response)

Reviewer #2: All comments have been addressed

2. Is the manuscript technically sound, and do the data support the conclusions?

Reviewer #1: Partly

Reviewer #2: Yes

3. Has the statistical analysis been performed appropriately and rigorously? 

Reviewer #1: I Don't Know

Reviewer #2: I Don't Know

4. Have the authors made all data underlying the findings in their manuscript fully available?

Reviewer #1: Yes

Reviewer #2: Yes

5. Is the manuscript presented in an intelligible fashion and written in standard English?

Reviewer #1: No

Reviewer #2: Yes

6. Review Comments to the Author

Reviewer #1: Dear authors,

thank you for addressing some of my concerns. The authors now acknowledge the limitations of their study, particularly the poor data basis.

Some points minor points remain however:

what does this sentence mean? Daily CRP infection? : Line 121-123 "Some authors even suggest the level of variability of CRP may influence clinical decisions, with one paper studying intensive care unit acquired infections showing daily CRP infection could be used as a marker of infection."

Last paragraph 375-379: From a clinical point of view: nobody will repeat a CRP test several times. This would be a strategy for a prospective study. In the clinic, based on the authors data, I would say one may assume a variation around a specific CRP value. Just remove the last part of the sentence behind the comma.

Reviewer #2: (No Response)

7. PLOS authors have the option to publish the peer review history of their article (what does this mean?). If published, this will include your full peer review and any attached files.

Reviewer #1: No

Reviewer #2: No

---

## [Editor Report · Acceptance letter]

4 Jun 2024

PONE-D-23-37309R1 

PLOS ONE

Dear Dr. Gough, 

I'm pleased to inform you that your manuscript has been deemed suitable for publication in PLOS ONE. Congratulations! Your manuscript is now being handed over to our production team.

Kind regards, 

on behalf of

Univ. Prof. Dr. Andreas Zirlik 

Academic Editor

PLOS ONE